



# Brief Communication: Time step dependence (and fixes) in Stokes simulations of calving ice shelves

Brandon Berg[1,2] and Jeremy Bassis[2]

[1]Physics Department, University of Michigan, Ann Arbor, Michigan, USA.
[2]Climate and Space Sciences and Engineering Department, University of Michigan, Ann Arbor, Michigan, USA.
**Correspondence:** Brandon Berg (brberg@umich.edu)

**Abstract.** The buoyancy boundary condition applied to floating portions of ice sheets and glaciers in Stokes models is numerically ill-posed when the glacier rapidly departs from hydrostatic equilibrium. This manifests in velocity solutions that diverge with decreasing time step size, contaminating diagnostic strain rate and stress fields. This can be especially problematic for models of calving glaciers, where rapid changes in geometry lead to configurations that depart from hydrostatic equilibrium and accurate measures of the stress field are needed. Here we show that the singular behavior can be cured with minimal computational cost by reintroducing a regularization that corresponds to the acceleration term in the stress balance. This regularization provides numerically stable velocity solutions for all time step sizes.

## 1 Introduction

Stokes simulations are used in glaciology as a tool to determine the time evolution of glaciers (e.g., Gagliardini et al., 2013). Increasingly, these models are also used to examine the stress field within glaciers to better understand factors that control crevasse formation and the onset of calving events (Ma et al., 2017; Benn et al., 2017; Nick et al., 2010; Todd and Christoffersen, 2014; Ma and Bassis, 2019). This type of model can provide insight into the relationship between calving, climate forcing and boundary conditions (e.g., Todd and Christoffersen, 2014; Ma et al., 2017; Ma and Bassis, 2019).

Here we show that a common method used to implement the ice-ocean boundary condition in Stokes models can result in solutions that are sensitive to the choice of simulation time step size. This behavior manifests in applications that allow for rapid changes in the model domain — a type of change associated with models that allow for instantaneous calving events or crevasses (Todd and Christoffersen, 2014; Todd et al., 2018; Yu et al., 2017).

The time step dependence arises because for glaciers outside of hydrostatic equilibrium, the acceleration is not small, as assumed in Stokes flow. We illustrate both the issue and the solution using an idealized ice shelf geometry (illustrated in Fig. 1), where the upper portion has calved away, emulating the "footloose" mechanism proposed by Wagner et al. (2014) where a aerial portion of the calving front first detaches.



## 2   Problem Description

### 2.1   Glacier Stress Balance

Denoting the velocity field by $\boldsymbol{u}(x,z,t) = (u_x(x,z,t), u_z(x,z,t))$ and pressure by $P(x,z,t)$, conservation of linear-momentum can be written in the form:

$$\nabla \cdot \boldsymbol{\sigma} + \rho_i \boldsymbol{g} = \rho_i \frac{D\boldsymbol{u}}{Dt}. \tag{1}$$

The Cauchy stress is defined in terms of strain rate, effective viscosity, pressure, and the identity matrix $I$:

$$\boldsymbol{\sigma} = 2\eta\boldsymbol{\varepsilon} - PI, \tag{2}$$

with strain rate tensor $\boldsymbol{\varepsilon}$ given by:

$$\varepsilon_{ij} = \frac{1}{2}\left(\frac{\partial u_i}{\partial r_j} + \frac{\partial u_j}{\partial r_i}\right). \tag{3}$$

Here $\rho_i$ is the density of ice, $\boldsymbol{g}$ is the acceleration due to gravity, and $\eta$ is the effective viscosity of ice:

$$\eta = \frac{B}{2}\varepsilon_e^{\frac{1}{n}-1}. \tag{4}$$

The effective viscosity is a function of the effective strain rate $\varepsilon_e = \sqrt{\varepsilon_{ij}\varepsilon_{ij}}/2$, a temperature dependent constant $B$, and the

flow-law exponent $n = 3$; the acceleration term on the right hand side of Eq. (1) denotes the material derivative.

In the Stokes limit, we drop the acceleration term from Eq. (1), an approximation which is justified for most glaciological applications (Greve and Blatter, 2009). However, as we shall show, this assumption is problematic for applications where the ice departs from hydrostatic equilibrium.

### 2.2   Boundary Conditions

To illustrate an example where the Stokes flow problem becomes ill-posed, we consider a two-dimensional floating ice shelf (Fig. 1). We specify the normal component of the velocity $\boldsymbol{u} \cdot \hat{n} = u_1$ where $u_1$ is a constant along the inflow portion of the domain ($\Gamma_1$). At the ice-atmosphere boundary ($\Gamma_2$) we assume the surface is traction free. At the boundary between ice and ocean ($\Gamma_3$) the shear traction along the ice-interface vanishes and continuity of normal traction along the ice-ocean interface can be written as $\sigma_{nn}(x,t) = -\rho_w g b(x,t)$ where $b(x,t)$ is the position of the ice-ocean interface.

Problems arise with this form if the glacier is not *exactly* in hydrostatic equilibrium because buoyancy forces along the ice-ocean interface cannot be balanced by internal stresses. In this case there is no solution and vertical velocities are singular. In reality, of course, the ice will quickly re-adjust to hydrostatic equilibrium through rapid buoyant uplift through the (nearly) inviscid ocean.

We can more accurately specify the boundary condition for Stokes flow at the ice-ocean interface by writing it in the form:

$\sigma_{nn}(x,t) = -\rho_w g\left(b(x,t) + \Delta z(x,t)\right)$    on   $\Gamma_3$ (5)





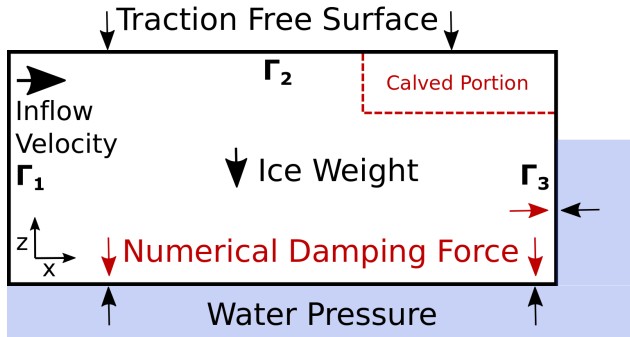

**Figure 1.** A diagram showing the boundary conditions of an idealized floating ice shelf. The ice-ocean interface is subject to two normal stresses - the depth dependent water pressure and the numerical damping force for stabilization to hydrostatic equilibrium. The dashed red line illustrates an iceberg that breaks off from the top of the calving front (exaggerated), reducing the freeboard and instantaneously perturbing the ice shelf from hydrostatic equilibrium.

where $\Delta z(x,t)$ is an a *priori* unknown uplift that must be determined as part of the solution to enable the full force balance to close.

The additional uplift term $\Delta z$ has a simple physical explanation: if normal stress was exactly hydrostatic, $\sigma_{nn} = \rho_i g H$ where $H$ is the ice shelf thickness. Equation (5) can then be solved for $\Delta z$ to determine the position of the bottom interface needed for the forces to balance. The Stokes limit is more complex as internal stresses also contribute to the normal stress at the ice-ocean interface, but the location of the ice-ocean interface needs to be solved for as part of solution to the problem, which we examine next.

### 2.3 Numerical Stabilization of Buoyant Uplift

Different numerical methods apply different techniques to solve for $\Delta z$ in Eq. (5). For example, in Elmer/Ice, a popular package for modeling Stokes glacier flow, Durand et al. (2009) proposed an ingenious solution in which $\Delta z$ is estimated based on a Taylor series of vertical position of the ice-ocean interface:

$$\Delta z = u_z(x,t)\Delta t + O(\Delta t^2). \tag{6}$$

This Taylor series transforms the uplift into a time step dependent Newtonian velocity damping:

$$\sigma_{nn}(x,t) = -\rho_w g \left( b(x,t) + u_z(x,t)\Delta t \right). \tag{7}$$

However, the coefficient of the damping force is proportional to $\Delta t$ and vanishes in the limit of small $\Delta t$. In this limit, vertical velocities are singular. We refer to the damping method given in Eq. (7) as the "sea-spring" method based on the nomenclature used in Elmer/Ice documentation.





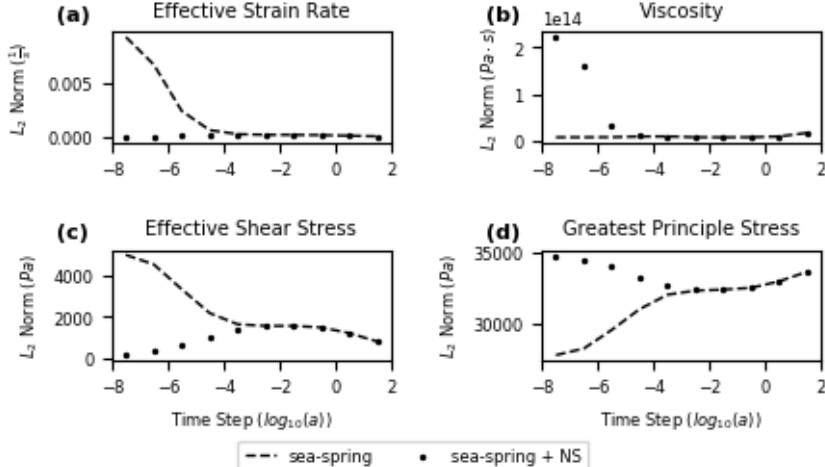

**Figure 2.** $L_2$ norm of **(a)** effective strain rate, **(b)** viscosity, **(c)** effective shear stress, and **(d)** greatest principle stress. Solutions shown for sea-spring damping and sea-spring with Navier Stokes (NS). For small time steps, the sea-spring solution diverges. The sea-spring with Navier Stokes term is well posed for all time steps and shows variability with time step.

With this method, we can decompose the velocity $\boldsymbol{u}$ into a "viscous" component $\boldsymbol{u}_{visc}$ and a hydrostatic uplift component $\boldsymbol{\Delta z}_{uplift}$, which we write in the form:

$$\boldsymbol{u}(\Delta t) = \boldsymbol{u}_{visc} + \frac{\boldsymbol{\Delta z}_{uplift}}{\Delta t}, \tag{8}$$

where $\boldsymbol{\Delta z}_{uplift}$ is the vector form of the same displacement written in Eq. (5). Due to the dependence of the ice-ocean boundary stress on time step size $\Delta t$, the total velocity becomes time step dependent.

Inspecting Eq. (8) shows that in the limit of large $\Delta t$, the uplift term becomes small compared to the viscous velocity. Thus, the sea-spring damping method can provide a good approximation for the viscous velocity as long as $(\boldsymbol{\Delta z}_{uplift})/(\boldsymbol{u}_{visc}\Delta t) <<$

1, which is true if the glacier is close to hydrostatic equilibrium ($\boldsymbol{\Delta z}_{uplift}$ small) or a sufficiently long time step is used.

## 3 Calving-Based Convergence Test

### 3.1 Test Design

For our test, we implement an idealized rectangular ice shelf of thickness 400 m and length 10 km. This ice shelf is initialized to be in exact hydrostatic equilibrium with inflow velocity for the upstream boundary condition set to $4\,\mathrm{km\,a^{-1}}$. These thickness

and velocity parameters are broadly consistent with observations for the last 10 km of Pine Island ice shelf (Rignot et al., 2017, 2011; Mouginot et al., 2012; Paden et al., 2010, updated 2018). The temperature dependent constant in Glen's flow law is chosen to be $1.4 \times 10^8\,\mathrm{Pa\,s^{\frac{1}{3}}}$, the value given by Cuffey and Paterson (2010) for $-10°$C.





To emulate the occurrence of a calving event that would perturb the ice shelf from hydrostatic equilibrium, a rectangular section of length 50 m and thickness 20 m is removed from the upper calving front of the glacier (Fig. 1). This type of calving behavior has been proposed as the trigger of a larger calving mechanism related to buoyant stresses on the ice shelf (Wagner et al., 2014). The numerical effects we document are not unique to this style of calving and this mechanism is only meant to illustrate the numerical issues.

The problem is implemented in FEniCS (Alnæs et al., 2015), an open source finite element solver with a Python interface that has been previously used for Stokes glacier modeling (Ma et al., 2017; Ma and Bassis, 2019). The problem is solved using mixed Taylor-Hood elements with quadratic elements for velocity and linear elements for pressure. The open source finite element mesh generator Gmsh is used to generate a unstructured mesh with uniform grid spacing of 10 m near the calved portion of the domain and grid spacing of 40 m elsewhere.

### 3.2 Divergent Behavior

Figure 2 shows the sensitivity of the velocity field, effective strain rate, effective shear stress, and greatest principle stress to time step size when using the sea-spring boundary condition from Eq. (7). Furthermore, because of the coupling between effective strain rate and viscosity, the viscosity for the majority of the domain becomes unphysically small as the time step decreases. This positive feedback between effective strain rate and viscosity is especially problematic as higher strain rate causes lower viscosity, and vice versa, leading to unphysical results. This problem can be alleviated by using a viscoelastic rheology when examination of short time scale behavior is desired. However, even for a purely viscous model, short time steps may be necessary to satisfy numerical stability criteria during hydrostatic adjustment that momentarily forces the model outside of the Stokes range.

In addition to the divergence of the $L_2$ norm, we also examine the maximum effective shear stress and greatest principle stress (Fig. 3). Maximum values may be a better predictor of the effect of time step dependence on the output of Stokes calving models. Because calving models often assume that calving is likely if a stress threshold is exceeded (Ma et al., 2017), outliers in stress are more important than a stress averaged over the entire domain.

## 4 Proposed Solution - Reintroduce Acceleration Term into Stress Balance

The velocity solution is ill-posed because the neglected acceleration term is not actually small relative to the other terms in Eq. (1): large velocities associated with hydrostatic adjustment rapidly change on time scales short compared to the internal deformation of the ice. We therefore reintroduce the acceleration term and show that this regularizes the solution for small time steps. We use a simple first order backwards differentiation scheme in a Lagrangian reference frame where

$$\frac{D\boldsymbol{u}_i}{Dt} = \frac{\boldsymbol{u}_i - \boldsymbol{u}_{i-1}}{\Delta t}. \tag{9}$$

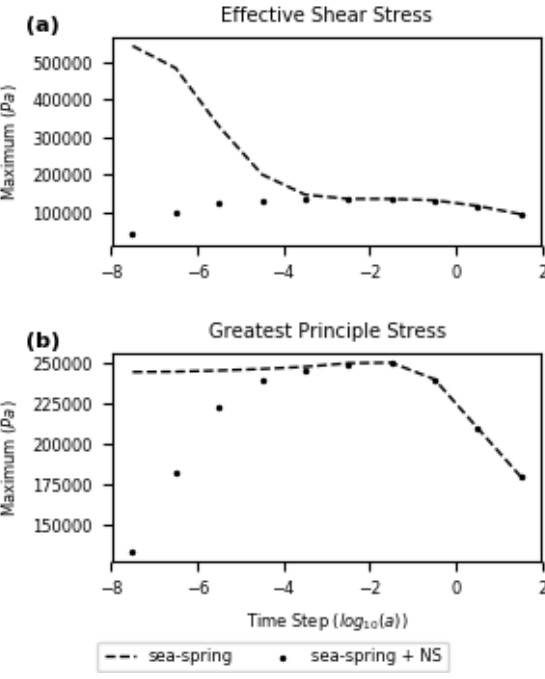

**Figure 3.** Maximum **(a)** effective shear stress and **(b)** greatest principle stress. Solutions shown for sea-spring damping and sea-spring with Navier Stokes (NS). Without corrections, maximum stress is overestimated in the Stokes model, which could lead to overestimation of glacier retreat due to calving.

This effectively introduces a Newtonian damping term on the entire body of the glacier where the damping coefficient $C = 1/\Delta t$. Computational difficulty is not impacted by reintroducing the acceleration term in this way because the term is
linear with respect to velocity. However, unless a fully implicit scheme was implemented, the solution becomes inaccurate (and unstable) for long time steps. Therefore, we propose to use both damping terms so that the system of equations is numerically accurate for small time step sizes and the velocity converges to the viscous limit for large time step sizes. Although this method rectifies the numerical inaccuracies present at short time steps with sea-spring damping, it does not address *physical* inaccuracies from using a rheology not suited for elastic effects. However, the numerical divergence exists independent of
rheology and would need to be addressed even for a viscoelastic model.

When we include both damping terms, effective strain rate, effective principle stress, and greatest principle stress are consistent for both small and large time steps (Fig. 2). At small time steps the acceleration term dominates, so the sea-spring with Navier-Stokes solution departs from the sea-spring solution. At large time steps, the sea-spring damping dominates and the two solutions overlap. At intermediate time steps, both damping terms contribute as the solution transitions from the regime
dominated by inertial effects to one where inertial effects are small. It is crucial to note that although the sea-spring with Navier-Stokes solution retains time step dependence for small time steps, the time step dependence now results from the

physical evolution of the system: the solution resolves the acceleration and deceleration of the glacier as it evolves towards a steady-state as opposed to being a consequence of an ill-posed problem.

Notably, the sea-spring solution shows larger maximum stresses than the sea-spring with Navier Stokes solution (Fig. 3).
This is particularly evident for the effective shear stress, which is overestimated by an order of magnitude at the shortest time step tested. In the footloose calving mechanism, when a portion of the upper calving front is removed, the front of the ice shelf becomes buoyant and produces increased shear stress upstream on the ice shelf (Wagner et al., 2014). This over prediction of stresses could cause a calving model to predict unphysical calving events due to numerical inaccuracies.

## 5   Conclusions

Our study shows that using a common numerical stabilization method of the ice-ocean boundary in Stokes glacier modeling there is an explicit time step dependence of the solution that diverges for small time steps when the domain departs from hydrostatic equilibrium. For model applications where changes in the domain are only due to viscous flow, the time step dependence is not problematic as long as domains are (nearly) in hydrostatic equilibrium at the start of simulation. However, for applications where rapid changes to the model domain occur, such as when calving rules are implemented, sudden departure
from hydrostatic equilibrium is not only possible, but expected. In these cases, time step dependence of the solution will appear. This can contaminate solutions of the stress after calving, potentially leading to a cascade of calving events and an overestimate of calving flux if numerical artifacts are not addressed. However, the time step dependence can be easily cured with little computational cost by reintroducing the acceleration term to the Stokes flow approximation. The acceleration term regularizes the solution for small time step sizes and results in a physically consistent solution.

*Author contributions.*   BB identified the numerical issue with guidance from JB. BB and JB developed the proposed solution to the numerical issue. BB prepared the manuscript with contributions from JB.

*Competing interests.*   The authors declare that they have no conflicts of interest.

*Acknowledgements.*   This work is from the DOMINOS project, a component of the International Thwaites Glacier Collaboration (ITGC). Support from National Science Foundation (NSF: Grant PLR 1738896) and Natural Environment Research Council (NERC: Grant NE/S006605/1).
Logistics provided by NSF-U.S. Antarctic Program and NERC-British Antarctic Survey. ITGC Contribution No. ITGC:010.





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
