# Peer review of "Brief Communication: Time step dependence (and fixes) in Stokes simulations of calving ice shelves"

_The Cryosphere, 2019_

## Referee Comment (RC1) · Anonymous Referee #1 · 2 Mar 2020

This paper presents a simple method to overcome time step dependence of the solution arising when solving for an ice-shelf which departs significantly for hydrostatic equilibrium. This could be the case for instantaneous non-vertical icebergs calving or supraglacial lake drainage.

This is quite a technical paper but as the problem might be encountered by other groups using different Stokes solver, this brief communication certainly deserve to be published. The overall writing of the paper is quite good even if I think that there is some room for improvement.
My main concern is the fact that the time step dependence of the solution is sometime seen as negative (e.g. title) or positive (e.g. caption Fig. 2). And indeed it is not completely clear from Figs. 2 or 3 to see which of the two solutions is the one that works better. The viscosity has no timestep dependence for the sea-spring solution and it is the sea-spring+NS solution that has no time step dependence for effective strain-rate. This is even less clear for stress where both solutions are diverging but presents both a timestep dependence. I would expect more comments on the text on this and how from the figure one can decide which is the working solution.

I have also listed a number of smaller points below.

- page 2, line 41: ", where $u_1$ is a constant"

- Figures 2 and 3: the quality of Figs. 2 and 3 are very low. It is not clear from the text and the captions if what is plotted on these figures is the solution after the timestep following the calving event. What are the differences of setup between Fig. 2 c and d and Fig. 3? I would suggest to modify Pa to MPa or kPa. For the x axis, the caption should tell that time step are varying from xx seconds to xx years?

- page 5, line 95: not sure the second sentence of part 3.2 should start with "Furthermore"?

- Eq. (9): specify that $u_{i-1}$ is the velocity at previous timestep?

- page 6, line 113: "where the damping coefficient is"

-

---

## Referee Comment (RC2) · Christian Schoof (Referee) · 25 Apr 2020

BRIEF COMMUNICATION: TIME STEP DEPENDENCE (AND FIXES) IN STOKES SIMULATIONS OF CALVING ICE SHELVES'

B. Berg and J.N. Bassis

**Overview**

Let me start with an apology to editor and authors: this has taken far too long to complete, global pandemics or otherwise.

This paper addresses how to adapt ice sheet / ice shelf flow models to situations in which overall force balance cannot be satisfied momentarily. Clearly this is not something that occurs for any extended periods of time in an ice sheet or ice shelf, but causes significant problems when it does: the Stokes flow equations that underpin most models for ice sheet and ice shelf flow have no solution when force balance is violated temporarily, since they omit inertial terms.

The obvious situation in which violation of force balance can occur is when an ice sheet that contains no grounded portion within the model domain experiences an abrupt calving event, removing part of the ice and causing an imbalance between buoyancy forces and weight of the ice. When I say 'contains no grounded portion', I really mean, 'is not subject to a Dirichlet or Robin condition constraining vertical velocity on any part of the boundary'; in the computational set-up in the paper, the left-hand boundary of a block of ice is 'attached' to an unmodelled ice shelf by prescribing horizontal velocity and vanishing vertical shear stress $\sigma_{zx}$; one could envisage ensuring force balance by altering the latter condition to account for some sort of vertical drag that is dependent on vertical velocity as a regularization (as a friction law, effectively), but that is beside the point.

The paper correctly (and importantly) makes the point that ice shelf models need to be cognizant of the fact that force balance violation is a real possibility, and that the omission of inertial terms leads to an ill-posed model. (I have a few things to add on ill-posedness shortly. There are different ways in which a model can be ill-posed and the distinction between them is important, in my opinion.) The authors also show that retaining the inertial term (specifically, the time derivative of velocity) fixes the problem. However, in order to maintain a stable time stepping scheme for large time steps, the retention of an additional stabilizing term due to Durand et al (2009) that mimics a fully implicit time step is necessary.

This is a useful contribution to the literature and should be published. I have a few comments on presentation and a few issues that could do with clarification, mostly at the authors' discretion. I apologize if I take some time describing these. It seems to be my habit to drone on about things that few others care about so feel free to ignore any or all of what follows,

1. The most important concept that could be clarified for the reader is the notion of ill-posedness, which also needs to be distinguished from the effective stiffness of

the problem that Durand et al (2009) were solving, which led them to invent their stabilizing 'sea spring' mechanism.

The problem in Durand et al's paper is perfectly well posed so long as there is contact with bedrock somewhere, and the normal stress exceeds water pressure on that contact area. In that case, a Dirichlet condition on normal velocity applies there, while the friction law provides a nonlinear Robin condition on tangential velocity (and hence on horizontal velocity if the bedrock boundary is not vertical everywhere). These conditions ensure mathematically that the Stokes flow problem has a unique solution that depends continuously on 'the data' (things like the boundary conditions), making the flow problem well-posed in the usual sense. There is an ample literature on well-posedness of the Stokes equations (going back to Ladzhenskaya in the 1960s) and the fact that pure stress conditions can lead to solvability conditions similar to the solvability condition for a pure Neumann conditions on a Poisson equation are well known in the pde literature, though I wouldn't be able to give you the 'original' citation for this. More on this shortly however.

The problem in Durand et al's work is that despite a formally continuous dependence on the data, the problem is extremely sensitive to slight deviations in vertical position relative to the one the ice shelf 'wants' to adopt. In other words, the problem is poorly conditioned or stiff, which is not quite the same as ill-posed.

Effectively, at leading order, Durand et al are modelling a long viscous beam, which permits extremely large vertical velocities if the lower ice surface does not adopt a very specific shape that is close to hydrostatically supported (though not exactly hydrostatically supported). Their real interest is not in solving the approach to that position, which requires very short time steps, but in modelling the much slower evolution of ice thickness due to horizontal flow. The 'sea spring' is indeed an ingenious way to dampen the vertical motions due to the the beam-like nature of the shelf (effectively, by providing the viscous analogue of a Winkler foundation in elastic beam theory).

Importantly, the sea spring is not intended to deal with a situation where force balance is violated, and I think it is important to make that point. I doubt whether the original authors had even considered that scenario.

Without Dirichlet or Robin conditions constraining both velocity components (keeping things in two dimensions throughout this discussion), between one and three solvability conditions arise for a Stokes flow problem. In the scenario outlined in the paper, horizontal velocity is constrained at the inflow boundary on the left, so the only solvability condition winds up being the one that says that the net vertical component of force on the shelf must be zero. If one were to replace the left-hand inflow boundary with another free 'cliff' on which only stress conditions apply analogously to the right-hand boundary, we would get an iceberg, subject to not one but two more solvability conditions: in that case, we would also need a

zero net horizontal component of force *and* a vanishing net torque.

Mathematically, this is tied up with the fact that the Stokes operator is invariant under the addition of a rigid body motion $\boldsymbol{r}$ (a combination of a constant velocity and a rotational velocity) to the velocity field $\boldsymbol{u}$, and the number of unconstrained degrees of freedom permitted by the boundary conditions is the number of solvability conditions that arise: in the example in the paper, the horizontal and rotational degrees of freedom in the rigid body motion are constrained by the Dirichlet condiition on on the horizontal velocity component, so we only get one solvability condition, associated with vertical force balance and corresponding to the vertical component of the constant part of $\boldsymbol{r}$.

As I mentioned already, there is an extensive literature that proves that net force and torque balance conditions that are not automatically taken care of by the boundary conditions provide not only necessary but sufficient conditions for the existence of solutions.

When these conditions are not satisfied (e.g., in the present paper, force balance in the vertical may not be satisfied), then well-posedness fails at the first hurdle: there is no solution at all. The sea spring does not solve this problem, as the authors discover: the solution to the Stokes flow problem with no intertial term and only the sea spring stabilizing boundary conditions becomes intrinsically dependent on time step size $\delta t$, and therefore has no continuum limit as $\delta t \to 0$. This is where something like the approach in the present paper is necessary.

There is a more subtle ill-posedness that the sea spring mechanism by itself *can* help mitigate. Ill-posedness requires not only existence, but also uniqueness. When force and torque balance conditions are satisfied, there is a solution, but it is not unique: you can in principle add any rigid body motion permitted by the boundary conditions and still obtain a solution (in the case described by the paper , the permitted rigid body motion is purely a vertical velocity, for an iceberg it would be an arbitrary rigid body motion).

In reality of course the motion of a chunk of ice is not indeterminate when force and torque balance are satisfied, and yet you do not need to appeal to intertial terms to figure out what the velocity field is (as the authors here point out, doing so on its own is a bad idea if you want to take long time steps, so retaining inertial terms may not even be a practical solution to the problem).

Given *a* solution $\boldsymbol{v}$ to the Stokes flow problem in which force and torque balance satisfied, you can figure out what rigid body motion you need to add by requiring that the displaced ice surface *after* the next time step is still such that force and torque balance are satisfied after that time step. Assuming for simplicity for the moment that a forward Euler step is used and that the boundary conditions on the boundary are in the form $\boldsymbol{\sigma} \cdot \boldsymbol{n} = \boldsymbol{f}_b(x, z)$ all along the boundary (so there are only Neumann conditions), this amounts to finding a rigid body motion $\boldsymbol{r} =$

$\boldsymbol{r}_0 + \omega(-z, -)$ with constant-in-space $\boldsymbol{r}_0$ an $\omega$ such that the domain boundary $\partial\Omega(t+\delta t)$ obtained by translating every point $\boldsymbol{X}(t) = (X(t), Z(t))$ on the boundary to the corresponding new position

$$\boldsymbol{X}(t + \delta t) = \boldsymbol{X}(t) + \boldsymbol{v}(\boldsymbol{X}(t))\delta t + \boldsymbol{r}_0\delta t + \omega(-Z(t), X(t))\delta t$$

satisfies

$$\int_{\Omega(t+\delta t)} \rho\boldsymbol{g}\,\mathrm{d}\Omega + \int_{\partial\Omega(t+\delta t)} \boldsymbol{f}_b\,\mathrm{d}\Gamma = 0$$

and

$$\int_{\Omega(t+\delta t)} \boldsymbol{x} \times \rho\boldsymbol{g}\,\mathrm{d}\Omega + \int_{\partial\Omega(t+\delta t)} \boldsymbol{x} \times \boldsymbol{f}_b\,\mathrm{d}\Gamma = \boldsymbol{0},$$

where $\boldsymbol{x} = (x, z)$ and $\Omega(t)$ is the ice domain at time $t$. This amounts to three conditions (the last is really a scalar condition for a two-dimensional domain), sufficient in principle to find the three constants that define $\boldsymbol{r}$: the two components of the constant vector $\boldsymbol{r}_0$ and the angular velocity $\omega$.[1] The case where the horizontal velocity is already constrained by a Dirichlet condition as in the paper works analogously.

The method proposed above is not quite the same as the sea spring (since there the constraint of force balance being satisfied after the next time step is built into the Stokes solver, rather than requiring a post-processing step to find the rigid body motion required to ensure force balance on the next time step), but the two approaches are close to each other.

**Recommendation:** I would make clear the distinction between the poorly condition Stokes flow problem in Durand et al and the two flavours of actual ill-posedness seen when your boundary conditions permit force and/or torque balance to fail. It won't hurt to allude to the latter, even though I don't imagine many people are trying to solve Stokes flow problems for icebergs — you never know. It would also be reasonable to say that the sea spring mechanism (probably) works well for the second version of the actually ill-posed case, where big departures from equlibrium need never occur.

2. As a sort of brief follow-on from the first point and the role of rigid body motions, this point concerns the discussion about the validity of the Stokes equations. If you use the non-dimensionalization procedure that is usually used to justify the Stokes equations from the Navier-Stokes equations for the case you are looking at here (failure of force balance, or potentially force and torque balance), you do not
* * *
[1]There is a caveat in the sense that the horizontal motion of an iceberg will remain indeterminate even with this additional constraint requiring force and torque balance after the next time step, because horizontal force balance will be satisfied trivially (identically) if the boundary force $\boldsymbol{f}_b$ is given by buoyancy, so an additional constraint such as zero mean velocity would be necessary.

really conclude that the Reynolds number has suddenly become $O(1)$ or large and therefore the full Navier-Stokes equations must be solved.

Instead, what you find is that you end up having to decompose the velocity field into two parts. I'll spare you the bit where I dress this up with mathematics and just describe what happens. The first part is a rigid body motion whose evolution is controlled by Newton's second law using the net force on the chunk of ice, and by the equivalent of Newton's second law for the evolution of angular momentum. The magnitude of this first part is much larger than the second part, so the *motion* of the chunk of ice as it settles into a new position in which force and torque balance are satisfied is simply that of a rigid block (intuitively obvious, i guess).

The second part is the solution of a *Stokes flow* problem, sans inertial terms, and this second part controls internal stresses during the settling process (if those are the main concern, which I think they are). Call this second part of the velocity the viscous velocity. The Stokes flow problem for the viscous velocity has the same boundary conditions as the original Navier-Stokes problem, but the body force is amended by subtracting the inertial terms generated by the rigid body motion. This ensures *apparent* force and torque balance in the problem for the viscous velocity, although if you want the viscous velocity to be unique (not particularly relevant since the motion is controlled by the rigid body motion and the viscous stresses are unique) you have to add something like requiring that the mean of the viscous velocity and the mean rotation due to the viscous velocity vanish.

In other words, you have

$$\boldsymbol{u} = \boldsymbol{r}(x, z, t) + \boldsymbol{v}(x, z, t)$$

with $|bmr$ a rigid body motion

$$\boldsymbol{r} = \boldsymbol{r}_0(t) + \omega(t)(-z, x), ,$$

where $\boldsymbol{r}_0$ and $\omega$ depend on time but not position. The ratio $|\boldsymbol{v}|/|\boldsymbol{r}|$ scales as $Re$, the Reynolds number that one would normally compute from the viscous velocity scale for the size of the applied body and surface forces. As a result a boundary point $\boldsymbol{X}(t)$ on $\partial\Omega(t)$ evolves as

$$\frac{\mathrm{d}\boldsymbol{X}}{\mathrm{d}t} = \boldsymbol{r}(\boldsymbol{X}(t)).$$

because $\boldsymbol{v}$ is tiny compared with $\boldsymbol{r}$: while force balance is violated, the ice domain moves as a rigid body. Assuming for simplicity again that stress $\boldsymbol{\sigma} \cdot \boldsymbol{n} = \boldsymbol{f}_b(x, z)$ is specified everywhere at the boundary, the rigid body motion satisfies

$$\int_{\Omega(t)} \rho \, \mathrm{d}V \frac{\mathrm{d}\boldsymbol{r}_0}{\mathrm{d}t} = \int_{\Omega} \rho\boldsymbol{g} \, \mathrm{d}V + \int_{\partial\Omega} \boldsymbol{f}_b \, \mathrm{d}\Gamma,$$

$$\frac{\mathrm{d}}{\mathrm{d}t} \int_{\Omega(t)} \rho\boldsymbol{x} \times \boldsymbol{r} \, \mathrm{d}V = \int_{\Omega} \boldsymbol{x} \times \rho\boldsymbol{g} \, \mathrm{d}V + \int_{\partial\Omega} \boldsymbol{x} \times \boldsymbol{f}_b \, \mathrm{d}\Gamma,$$

where the second equation can be cast in terms of the sum of a moment of intertia (that is constant due to the volume $\Omega(t)$ moving as a rigid body) times $\mathrm{d}\omega/\mathrm{d}t$ and the time derivative of the moment of inertia associated with the barycenter of $\Omega(t)$, which can be written in terms of the time derivatives of $\boldsymbol{r}_0$ and $\omega$. In short, the current position and orientation of the rigid body $\Omega(t)$ uniquely defines the derivatives of $\boldsymbol{r}_0$ and $\omega$, while $\boldsymbol{r}_0$ and $\omega$ determine the change of position and orientation of $\Omega(t)$ — basic non-continuum mechanics, if you will.

The viscous velocity $\boldsymbol{v}$ by contrast satisfies the modified Stokes flow problem

$$\nabla \cdot \boldsymbol{v} = 0$$

$$\boldsymbol{0} = \nabla \cdot \boldsymbol{\tau}(\boldsymbol{v}) - \nabla p + \rho \boldsymbol{g} - \rho \left( \frac{\partial \boldsymbol{r}}{\partial t} + \boldsymbol{r} \cdot \nabla \boldsymbol{r} \right)$$

on $\Omega(t)$, subject to $(\boldsymbol{\tau}(\boldsymbol{v}) - p\boldsymbol{I}) \cdot \boldsymbol{n} = \boldsymbol{f}_b$ on $\partial\Omega(t)$, where $\boldsymbol{\tau}(\boldsymbol{v})$ is the viscous relationship between deviatoric stress and velocity, $p$ is pressure and $\boldsymbol{I}$ the identity tensor. The construction here ensures that the relevant force and torque balance relationships for $\boldsymbol{v}$ are automatically satisfied, while $\boldsymbol{r}$ can be solved for a priori, so the fictitious force term in the momentum balance equation is known before $\boldsymbol{v}$ is solved for.

This is *similar* to what is alluded to in equation (8) in the paper (more on that below), but not the same, since you can effectively solve for the rigid body motion $(\Delta z_{uplift}/\Delta t)$ and the viscous velocity separately, but that is not how the algorithm in the paper works. I just think it is worth pointing out somewhere that a construction of this kind is possible, which retains the idea of a Stokes flow describing the stress even as inertial effects kick in: in particular, the inertial motion is independent of rheology and viscous stresses, so long as the latter do not cause further domain changes by *fracturing* (they do not by causing viscous deformation of the ice body, that is far too slow, see the point about the ratio between the velocity magnitudes above).

As a side note, I think equation (8) is misleading if one were ever to try to allow rotational inertial motions, more on that under 'minor points' — the clue is in the form of the fictitious force term above.

**Recommendation**: The paper says 'However, as we shall show, this assumption is problematic for applications where the ice departs from hydrostatic equilibrium.' I'd circle back to this at some point and point out that things may not be quite so dramatic as to say that the Stokes equations have nothing to say about what happens during these 'inertial' events; they do, but in modified form. I think this will also tie in to the discussion of how to formulate the inertial term in discrete form, see again under 'minor points' below.

**Minor points**

- I would probably make a bit clearer how boundary conditions are important in determining whether an actual ill-posedness can occur in the sense of there being no solution to the Stokes flow problem. As the extended discussion above indicates, the partial Dirichlet conditions in the present paper ensure there is no issue of torque or horizontal force imbalance, but it others may run into these issues in their own research, and look to apply the method developed here. Also, as indicated in the second paragraph of this review, there may be other tricks to ensuring force balance.

- The decomposition in equation (8): for a sea spring model, I don't think you can argue that the sea spring term $\boldsymbol{u}(x,z)\delta t$ simply causes an additive term $\Delta z_{uplift}/\delta t$ As a simpler example, consider a Neumann condition in a Poisson equation

$$-\nabla^2 u = f \qquad\qquad\qquad \text{on } \Omega$$
$$\frac{\partial u}{\partial n} = g_n \qquad \text{on } \partial\Omega$$

and call the solution of this problem $u_{visc}$. Then modify (regularize?) the boundary condition as
$$\frac{\partial u}{\partial n} + cu\delta t = g_n$$

To the best of my knowledge, the solution to the modified problem cannot be written (as a function of $\delta t$!) in the form

$$u = u_{visc} + \frac{u'}{\delta t},$$

which is effectively what equation (8) is saying, albeit for a more complicated elliptic problem.

- Notation: there is quite a bit of randomness about which quantities are in boldface and which are not, especially when it comes to tensors ($\boldsymbol{\sigma}$ versus $\varepsilon$ and $I$?). Make it consistent to please the eye...

- The '/2' should probably be inside the square root in the definition of the invariant $\varepsilon_e$ just after equation (4)

- Writing $\boldsymbol{u}(\Delta t)$ on the left-hand side of (8) is confusing as $\boldsymbol{u}$ has a well-defined meaning as the continuum solution of the Navier-Stokes problem as a function of $(x, z, t)$, so changing the arguments of that function haphazardly to $\Delta t$ is bad form (and actually confused me quite a bit). For starters, the quantity on the left isn't $\boldsymbol{u}$ but a numerical approximation to it, solving a modified problem, so give it a different symbol, and be clear why you are using the $\Delta t$ argument on the left (your numerical algorithm thus constructed leads to a solution that turns out to depend on $\Delta t$, whereas you would want it not to be dependent on $\Delta t$.

- The numerical form of the acceleration term in equation (9): this is defensible when you have no rotational degrees of freedom in the rigid body motion, because the advection term for momentum $\boldsymbol{u} \cdot \nabla \boldsymbol{u}$ is dominated by $\boldsymbol{r} \cdot \nabla \boldsymbol{r}$ (see point 2 above), and in the absence of a rotational degree of freedom, $\nabla \boldsymbol{r} = \boldsymbol{0}$ so the advection term goes away. As soon as there is rotation, this is no longer true, and you are well-advised to retain the full inertial term $\partial \boldsymbol{u}/\partial t + \boldsymbol{u} \cdot \nabla \boldsymbol{u}$. I realize that the present paper does not allow for that possibility, but I think it is worth mentioning.

- Again, equation (9): I am actually not clear how you imagine you are computing this in a 'Lagrangian frame' to begin with, since you are solving, in discrete terms, an elliptic equation (or a parabolic equation with a backward Euler step, which is the same thing); if you genuinely are using a Lagrangian transformation here, please be explicit and speciifc. In terms of implementation, the reduced acceleration term in equation (9) is my biggest concern, even if I believe it to be leading-order correct (in the Reynolds number, see above) for the vertical-motion-only case discussed in the paper.

Christian Schoof, University of British Columbia

---

## Author Comment (AC1) · 29 May 2020

Responses to Referee comments by Anonymous Referee #1 on "Brief communication: Time step dependence (and fixes) in Stokes simulations of calving ice shelves" by Brandon Berg and Jeremy Bassis

We thank the anonymous reviewer for their feedback on this manuscript. Our responses to comments are given below, with original comments in black and responses in red. When referenced, line numbers refer to the revised manuscript.

**General Comments:**

This paper presents a simple method to overcome time step dependence of the solution arising when solving for an ice-shelf which departs significantly for hydrostatic equilibrium. This could be the case for instantaneous non-vertical icebergs calving or supraglacial lake drainage. This is quite a technical paper but as the problem might be encountered by other groups using different Stokes solvers, this brief communication certainly deserves to be published. The overall writing of the paper is quite good even if I think that there is some room for improvement.

We thank the reviewer for their positive comments. We respond in more detail to each comment below.

My main concern is the fact that the time step dependence of the solution is sometimes seen as negative (e.g. title) or positive (e.g. caption Fig. 2). And indeed it is not completely clear from Figs. 2 or 3 to see which of the two solutions is the one that works better.

This is a good point and we agree that the original figures were confusing. We have added text to the Figure 2 caption clarifying that the time step variability for the sea-spring with Navier Stokes solution is connected to the time evolution of the *system*. In the classic Stokes system, the velocity depends solely on the geometry of the ice shelf/glacier and internal properties (e.g., temperature). Hence, the dependence of the velocity field on time step is unphysical. However, when solving the full Navier-Stokes system, the velocity becomes time dependent and, like all numerical ODE integrations, we must take sufficiently small time steps to ensure numerical convergence when integrating with respect to time to find the numerical approximation to the solution. For the sea-spring + NS solution, the velocity tends towards zero as time step size decreases. As a consequence there is no deformation and, for very small times, the ice shelf behaves as a nearly rigid body as it approaches hydrostatic equilibrium. In the sea-spring + NS method, taking small time steps allows us to resolve the quasi-rigid body uplift of the ice shelf as it "bobs" in the water. We have added text in Section 4 of the manuscript clarifying the rigid body behavior at short time steps (lines 132-135).

This is illustrated below, where we show a plot of the L2 norm of the greatest principal stress for the sea-spring + NS method. The points represent different time step sizes (as in the manuscript). The dashed line is created by choosing the smallest time step and numerically integrating the system in time. For small times, the solution obtained from taking a single step with different time step size and the solution obtained from numerical integration (with a very small time step size) are similar, but begin to differ as time step size increases because taking large time steps results in less accurate solutions. Thus, the variation from the sea-spring + NS solution at small time steps is consistent with the actual time evolution of the system. This is connected to our discussion in section 4 of the manuscript.

[Figure]

The viscosity has no timestep dependence for the sea-spring solution and it is the sea-spring+NS solution that has no time step dependence for effective strain-rate.

We have chosen to omit plotting the viscosity form the final manuscript, instead replacing it with a plot of vertical velocity because effective strain rate and viscosity display somewhat redundant information and to ease the exposition of this short manuscript. Showing the vertical velocity directly provides a more direct illustration of the problem with the vertical velocity becoming unphysically large as time step size decreases.

However, we include a plot of viscosity below that shows the minimum and maximum value of the viscosity at different time steps for the two methods. Plotting the maximum and minimum values better highlights the time step dependence of the viscosity at small time steps. Note that for small time step sizes (and times) with the sea-spring + NS solution, the motion is quasi-rigid (i.e., deformation is small), strain rates are small and the viscosity increases. The viscosity is ultimately limited by the regularization we use for the rheology at small strain rates. By contrast, for the sea-spring solution, as time step size decreases the increasingly large vertical velocity causes a large strain rate increase. Because the viscosity and strain rate are inversely related, this results in decreasing viscosities.

[Figure]

This is even less clear for stress where both solutions are diverging but presents both a timestep dependence. I would expect more comments on the text on this and how from the figure one can decide which is the working solution.

We have clarified our language and removed most uses of the word "divergent" from the manuscript. The vertical velocity, which we now show in Figure 2 panel (a), becomes unphysically large for small time step sizes for the sea-spring method.  Effective strain rate becomes approximately zero for small time step sizes for the sea-spring + NS method because the motion at small times is nearly rigid body. Effective shear stress tends to zero for the sea-spring + NS method for the same reason as the effective strain rate. We also point out in section 4 that while Figure 3 shows higher maximum stresses for the sea-spring method, the L2 norm shows higher stresses for the sea-spring + NS method because of high negative compressive stresses (lines 136-137).

**Smaller Points:**

page 2, line 41: ", where u1 is a constant"

Comma has been added.

Figures 2 and 3: the quality of Figs. 2 and 3 are very low.

Figures have been changed from .png to .eps to improve viewing quality.

It is not clear from the text and the captions if what is plotted on these figures is the solution after the timestep following the calving event.

Thanks for pointing this out. We now state in the figure captions that the plotted solution is immediately after the (emulated) calving event.

What are the differences of setup between Fig. 2 c and d and Fig. 3?

The difference between Fig. 2 c and d and Fig. 3 is that we plot the L2 norm of the solution in Figure 2 and the L1 norm (maximum) of the solution in Figure 3. This is emphasized in the text and on figure y-axes. We plot the L1 norm (maximum) because this is the criterion that is often used in stress based calving laws, like the Nye zero stress. We show the L2 norm because it is (often) a more robust diagnostic of the behavior of the numerical solution.

I would suggest to modify Pa to MPa or kPa. For the x axis, the caption should tell that time step are varying from xx seconds to xx years?

We have modified the axes for stress to be kPa rather than Pa. Figure caption now states that the time step ranges from 1 second to 30 years.

page 5, line 95: not sure the second sentence of part 3.2 should start with "Furthermore"?

Thanks. "Furthermore" has been removed.

Eq. (9): specify that $u_{i-1}$ is the velocity at previous timestep?

A sentence has been added after the equation defining the variable.

page 6, line 113: "where the damping coefficient is"

Correction has been made.

---

## Author Comment (AC2) · 29 May 2020

Responses to Referee comments by Christian Schoof on "Brief communication: Time step dependence (and fixes) in Stokes simulations of calving ice shelves" by Brandon Berg and Jeremy Bassis

We thank Christian Schoof for his feedback on this manuscript. Our responses to comments are given below, with original recommendations/points in black and responses in red. When referenced, line numbers refer to the revised manuscript.

**Major Recommendations:**

We thank the reviewer for his comments and suggestions. We have incorporated most of the reviewers' excellent suggestions in the manuscript. We have vacillated slightly in our preferred terminology between ill-posed and stiff before settling on "unphysical" for reasons that are described in more detail in response to specific reviewer comments.

I would make clear the distinction between the poorly condition Stokes flow problem in Durand et al and the two flavours of actual ill-posedness seen when your boundary conditions permit force and/or torque balance to fail. It won't hurt to allude to the latter, even though I don't imagine many people are trying to solve Stokes flow problems for icebergs — you never know. It would also be reasonable to say that the sea spring mechanism (probably) works well for the second version of the actually ill-posed case, where big departures from equilibrium need never occur.

This is a good point. Our emphasis here was really on pointing out that when the geometry evolves rapidly, as is the case for an iceberg calving event, the sea-spring method becomes problematic and can lead to numerical problems. And when these numerical inaccuracies are combined with, for example, stress based calving criteria, there is the possibility of introducing purely numerical calving instabilities. However, under most circumstances, the sea-spring method remains satisfactory. This is better emphasized in lines 69-76.

We have added additional text to clarify the fact that, for our choice of boundary conditions, global force and torque balance are not necessarily satisfied leading to an ill-posed problem (lines 54-56, lines 110-114). However, if we consider fixed velocity (Dirichlet) boundary conditions over a portion of the domain, there is no rigid body motion (translation or rotation) that can be added to the ice shelf. In this case we can still obtain large velocities that are time step dependent using the sea-spring method when the geometry departs significantly from hydrostatic equilibrium over a portion of the domain.

Starting with a geometry that exactly satisfies global and local force/torque balance and then introducing small changes to that geometry can result in large changes to the velocity. This is what we think the reviewer calls "stiff" or poorly conditioned. Small changes in the initial conditions (i.e. position of the ice water interface) lead to large changes in the velocity solution. This can be partly cured by adding, say, a quadratic drag force due to the water, as the reviewer notes. However, for realistic drag coefficients, this still results in exceptionally large velocities. In fact, for configurations that we tested, the velocities can exceed the speed of sound! Because of this and because of the fact that we wish to avoid any confusion between "ill-posed", "ill-conditioned", and "stiff", we have decided to rephrase and call this behavior "unphysical". We believe this captures the numerical issue accurately and avoids introducing additional jargon that glaciologists might not be as familiar with.

The paper says 'However, as we shall show, this assumption is problematic for applications where the ice departs from hydrostatic equilibrium.' I'd circle back to this at some point and point out that things may not be quite so dramatic as to say that the Stokes equations have nothing to say about what happens during these 'inertial' events; they do, but in modified form. I think this will also tie in to the discussion of how to formulate the inertial term in discrete form, see again under 'minor points' below.

As stated above, if we consider fixed velocity (Dirichlet) boundary conditions over a portion of the domain, there is no rigid body motion (translation or rotation) that can be added to the ice shelf and we still obtain large velocities that are time step dependent using the sea-spring method. The large velocities are tied to the bending of the ice that occurs in response to removal of ice at the calving front. As the reviewer notes, this is because the problem is "stiff".

But to simplify our discussion, we have removed equation (8) and the accompanying text regarding the separation of the velocity into viscous and uplift components. Instead, we focus on highlighting the "stiffness" of the problem and how small changes to the ice-ocean boundary location can cause large changes in the solution. In this way, we emphasize the importance of carefully treating the hydrostatic uplift without commenting on the exact nature of the decomposition of viscous and rigid body motion. However, we do add text in the manuscript stating that such a decomposition may be possible (lines 110-114).

**Minor Points:**

I would probably make a bit clearer how boundary conditions are important in determining whether an actual ill-posedness can occur in the sense of there being no solution to the Stokes flow problem. As the extended discussion above indicates, the partial Dirichlet conditions in the present paper ensure there is no issue of torque or horizontal force imbalance, but it others may run into these issues in their own research, and look to apply the method developed here. Also, as indicated in the second paragraph of this review, there may be other tricks to ensuring force balance.

We have added text to clarify this. In particular, we have noted that adding global constraints on force and torque balance is possible (lines 110-114). We do, however, note that because ice breaks, global force and torque balance would have to be considered on each intact segment and this becomes increasingly challenging to efficiently identify and manage. Hence, our solution of simply including the acceleration directly into the Stokes equations becomes a more practical solution. As noted in our previous response, we have also shifted our terminology to "unphysical" because unphysically large velocities are still possible when drag is included.

The decomposition in equation (8): for a sea spring model, I don't think you can argue that the sea spring term  $u(x, z)\delta t$  simply causes an additive term  $\Delta z$ uplift/ $\delta t$ ...

We have streamlined and simplified this section and have eliminated this equation and explanation. We now focus on the "stiff" nature of the problem rather than a specific decomposition into viscous and uplift components.

Notation: there is quite a bit of randomness about which quantities are in boldface and which are not, especially when it comes to tensors ( $\sigma$  versus  $\epsilon$  and I?). Make it consistent to please the eye. . .

**Notation has been changed so that both vectors and tensors are all in boldface.**

The '/2' should probably be inside the square root in the definition of the invariant  $\varepsilon$  just after equation (4)

**Error has been fixed.**

Writing  $u(\Delta t)$  on the left-hand side of (8) is confusing as u has a well-defined meaning as the continuum solution of the Navier-Stokes problem as a function of (x, z, t), so changing the arguments of that function haphazardly to  $\Delta t$  is bad form (and actually confused me quite a bit). For starters, the quantity on the left isn't u but a numerical approximation to it, solving a modified problem, so give it a different symbol, and be clear why you are using the  $\Delta t$  argument on the left (your numerical algorithm thus constructed leads to a solution that turns out to depend on  $\Delta t$ , whereas you would want it not to be dependent on  $\Delta t$ .

**We have eliminated Equation (8) in response to other comments by the reviewer.**

The numerical form of the acceleration term in equation (9): this is defensible when you have no rotational degrees of freedom in the rigid body motion, because the advection term for momentum  $u \cdot \nabla u$  is dominated by  $r \cdot \nabla r$  (see point 2 above), and in the absence of a rotational degree of freedom,  $\nabla r = 0$  so the advection term goes away. As soon as there is rotation, this is no longer true, and you are wel advised to retain the full inertial term  $\partial u/\partial t + u \cdot \nabla u$ . I realize that the present paper does not allow for that possibility, but I think it is worth mentioning.

In our model, we are using an Arbitrary Langrangian Eulerian (ALE) formulation, which updates all mesh coordinates at every time step based on the velocity field. This led us to use the material derivative in Equation (9). But we have added text to clarify the fact that even in a Eulerian reference frame we could neglect the  $u \cdot \nabla u$  term because the velocity field does not contain a rigid body rotation (lines 117-118). We have also added text explicitly stating we are using an ALE formulation (lines 90-92).

Again, equation (9): I am actually not clear how you imagine you are computing this in a 'Lagrangian frame' to begin with, since you are solving, in discrete terms, an elliptic equation (or a parabolic equation with a backward Euler step, which is the same thing); if you genuinely are using a Lagrangian transformation here, please be explicit and specific. In terms of implementation, the reduced acceleration term in equation (9) is my biggest concern, even if I believe it to be leading-order correct (in the Reynolds number, see above) for the verticalmotion-only case discussed in the paper.

We are not quite sure that we understand the reviewer's question. In our implementation, we are solving a parabolic equation with a backward Euler step. The update to the ice geometry is done using a fully Langrangian formulation in which we update the mesh coordinates at every time step. Of course, we do need an initial condition for particle velocities. For the initial condition on velocity to use in the backward Euler step, we choose a uniform velocity field equal to the inflow velocity in the horizontal direction and zero in the vertical direction. The choice of zero initial velocity in the vertical direction is motivated by the experimental design, in which we

imagine an ice shelf that is initially perfectly at hydrostatic equilibrium, and thus should have nearly zero vertical velocity before calving. We have added text to section 4 of the manuscript to clarify the precise initial condition on velocity (lines 118-121).

---

## Author Response (AR2)

Responses to second round of Referee comments by Christian Schoof on "Brief communication: Time step dependence (and fixes) in Stokes simulations of calving ice shelves" by Brandon Berg and Jeremy Bassis

We thank Christian Schoof for his feedback on this manuscript. Our responses to comments are given below, with original recommendations/points in black and responses in red. When referenced, line numbers refer to the revised manuscript.

The revised paper is very well written and addresses the points that I raised. There are a few points of clarification that I would like to see, but i don't need to re-review the paper.

We thank the referee for his positive feedback and explain how we addressed each comment below.

1. In stating the boundary conditions, I think it's important to state the second boundary condition on Gamma_1. I think this must be vanishing shear stress sigma_zx; if the left-hand end of the domain was clamped in the sense of horizontal *and* vertical velocity being fixed, the solution would be unique and force balance would be ensured

Good point. We have added text to the boundary conditions section (2.2) of the text (Line 41) to state that shear traction also vanishes along the inflow boundary.

2. In the plots in figure 2, it is still ambiguous what is being plotted. I can compute maximum principal stresses, L2 norms of stress etc *instantaneously* after the removal of the chunk of ice that causes the readjustment of the orientation of the ice block in the water, (or at any other fixed point in time after that removal), I can average over a given period of time, or I can take the maximum of the quantities indicated over a given period of time (like (0,infty)). The last of these would make the most sense as otherwise it's a bit of a case of comparing apples and oranges. Please be specific.

We have added an additional sentence to section 3 (Line 92) to state more specifically that we are plotting values computed "immediately" after the removal of the chunk of ice. Previously we had stated this in the Figure captions but it was not present in the main body text. We have also added text to Figure captions explicitly stating that plotted values are calculated after a single time step to illustrate the fact that the stress depends sensitively on the chosen time step size.

3. The manuscript still dances around the fact that there is a true solution and a fake solution. The "fake" solution is the sea spring without the intertial terms, which can never be meaningful since it relies on a numerical stabilizer to ensure a solution in a situation where there is none. The true solution is the solution with the inertial term in place, computed for small dts. The large-dt solutions for either method are useful if you only care about flow, but not about the transient stress field. That should be made clearer (this was my point about ill-posedness in my first review; the new manuscript says "singular velocities" which is slightly misleading as what it

should really say is "there is no solution" since "singular velocity" suggests a local infinity). It's not so much that the method suggested in the paper is *better* than the sea spring for computing transient stresses, its the only way to compute transient stresses following the abrupt removal of a portion of ice.

One particular sentence I'd alter in that regard is ". However, even for a purely viscous model, short time
steps may be necessary to satisfy numerical stability criteria during hydrostatic adjustment that momentarily forces the model outside of the Stokes range." - you've already said that the underlying problem sans the sea spring has no solution (or at least implied it), why not just say that, in this particular configuration, the sea spring method has a solution only because of the regularizing term, but that solution diverges as dt -> 0, and is therefore meaningless. (I'd note that the"small time steps may be necessary" portion doesn't seem to be borne out computationally, as in, if you really don't care about whether you get transient stresses right: both versions of the model, with or without inertial terms, seem to compute just fine with large dt, so long as the inertial model has a sea spring mechanism)

This is a good point.  The mention of "singular velocities" has been removed from the manuscript and replaced with statements indicating that there is no solution to the problem.

Our original intention with this sentence was to highlight the fact that if someone wanted to accurately compute the full transient evolution, including the bobbing up and down of the ice shelf/berg, then time step sensitivity studies would be needed to ensure a numerically accurate solution.  (Just because the solution is stable doesn't mean it is accurate!).  But we agree with the reviewer that this statement is confusing and have removed it from the manuscript along with the preceding sentences mentioning different rheologies.

4. This passage is a bit odd:" The unphysically large velocities at small time
steps are magnified by the coupling between effective strain rate and viscosity. As effective strain rates become larger with small time steps, viscosity decreases, causing even greater strain rates." Magnified relative to what? Newtonian ice?

Good point.  We removed this passage rom the manuscript. See also our response to comment 3.

5. Following on from 4: "This problem can be alleviated by using a viscoelastic rheology when examination of short time scale behavior is desired. " - it's not clear to me that this isn't speculation, or what it really means. If by contrast we're looking at this still in the contrast of the non-inertial sea spring model, then the mere change of rheology without introducing inertial terms won't change the fact that the underlying solution is meaningless because force balance is violated. If instead this is about the need to change rheologies because a viscous model is inappropriate at the short time scales over which the shelf orientation adjusts to the removal of an ice block, then this is better discussed once the need to include inertial terms has been firmly

established (ie after the next subsection". In that case, however, I'm not sure it's easy to make statements about what the change in rheology implies for stresses; after all we now have to worry about elastic waves resulting from the removal of the ice block, and have to take seriously short time steps.. (This seems more like the domain of the sorts of discrete element models used in e.g. Bassis and Jacobs, or HiDEM),

Also a good point.  We removed this to avoid any unnecessary confusion.

5. I think it's "principal stress" not "principle stress" throughout. I don't think "principle" is ever used as an adjective.

That was embarrassing. Thanks for pointing this out. It has been corrected to be consistent throughout the manuscript.

[revised manuscript text omitted]